# Reported sleep duration reveals segmentation of the adult life-course into three phases

A. Coutrot [1] ✉, A. S. Lazar[2], M. Richards[3], E. Manley[4], J. M. Wiener [5],
R. C. Dalton [6], M. Hornberger[2] ✉ & H. J. Spiers [7] ✉

Classically the human life-course is characterized by youth, middle age and old age. A wide range of biological, health and cognitive functions vary across this life-course. Here, using reported sleep duration from 730,187 participants across 63 countries, we find three distinct phases in the adult human life-course: early adulthood (19-33yrs), mid-adulthood (34-53yrs), and late adulthood (54+yrs). They appear stable across culture, gender, education and other demographics. During the third phase, where self-reported sleep duration increases with age, cognitive performance, as measured by spatial navigation, was found to have an inverted u-shape relationship with reported sleep duration: optimal performance peaks at 7 hours reported sleep. World-wide self-reported sleep duration patterns are geographically clustered, and are associated with economy, culture, and latitude.

Sleep duration substantially varies within and between individuals. Understanding the determinants of these variations is key in many health and social domains, as sleep is essential for our well-being[1]. Sleep has a profound effect on our bodily functions ranging from the way our genes and cells operate[2] through various physiological processes, from immunity and metabolism[3] to complex brain functions involving cognition and mental health[4–6]. Both genetic (e.g., sex) and environmental (e.g., artificial lighting) factors modulate sleep[7,8]. In particular, age has been shown to explain a large share of the variance in sleep duration within the general population, with children sleeping considerably longer and better than adults, and younger adults sleeping less than older ones[9]. Reported sleep duration has been found to vary across nations, with Asians (e.g., Japan, Indonesia, Malaysia, Philippines) sleeping on average less than other nations[10–12]. However, few epidemiological studies have reported sleep patterns in the general population across nations. They have often focused on a particular age group[13], clinical population[14], or studied a specific sleep problem[15]. Previous studies also often relied on modest sample sizes (<1000 individuals), and when a larger sample was reported it consisted of aggregate data across studies with heterogeneous designs[9,10,16]. Furthermore, sleep epidemiology has almost exclusively focused on

high-income countries[17]. Since many sleep-related environmental factors substantially vary with countries' level of development (e.g., paid work time, artificial lighting), the results reported in the literature may not generalize to low- and middle-income countries (84% of the world's population).

Here, we present the distribution of reported sleep durations of 730,187 participants spread over 63 countries. This data was collected as part of the Sea Hero Quest project, a mobile video game designed to assess navigation ability in the global population[18–20]. Navigation ability is a multifaceted construct involving several cognitive processes, and has been strongly associated with many life outcomes such as academic achievement and career success in science[21]. Participants were asked several questions including: How much sleep do you get on average each night? A total of 3,881,449 participants played at least one level of the game, and 27.6% provided their average sleep duration. We removed participants above 70 years old because we have previously shown a strong selection bias in participants above this age, causing their performance to be substantially higher than would be expected in unselected participants of the same age[18]. This selection bias can be due to the fact that internet and mobile devices adoption rate are particularly low in less educated older adults[22]. As a result,

[1]LIRIS—CNRS—University of Lyon, Lyon, France. [2]Norwich Medical School, University of East Anglia, Norwich, UK. [3]Unit for Lifelong Health and Ageing, University College London, London, UK. [4]School of Geography, University of Leeds, Leeds, UK. [5]Department of Psychology, Bournemouth University, Poole, UK. [6]School of Architecture, Lancaster University, Lancaster, UK. [7]Institute of Behavioural Neuroscience, University College London, London, UK. ✉e-mail: antoine.coutrot@cnrs.fr; M.Hornberger@uea.ac.uk; h.spiers@ucl.ac.uk

older participants are less likely to be representative of their class of age. We removed participants with an average reported sleep duration above 10 h (0.6%) or below 5 h (0.9%). To ensure robust findings within countries we removed participants from countries with fewer than 500 players. This resulted in 730,187 participants from 63 countries included in our analysis, (see Table S1).

## Results

### Distribution of reported sleep duration in the general population

The world average declared sleep duration was 7.01h (SD = 1.07h, Fig. 1a). women reported sleeping 7.5 min more than men on average (Hedge's $g = 0.12$, 95%CI = [0.11 0.12]), see Fig. 1b. Figure 1c shows reported sleep duration as a function of age, with three distinct phases present across the life-course. First, a sharp decrease from early adulthood (women = 7.4 h, men = 7.3 h) to 35 y.o. (women = 7.0 h, men = 6.8 h). Then, the decrease slows down and plateaus until 50 years old (women = 6.9 h, men = 6.8 h). Finally, the average reported sleep duration increases again until 70 y.o., where it reaches the same value as at 30 y.o. for men and at 25 y.o. for women (women = 7.0 h, men = 7.1 h). The precise change points (33 y.o. and 53 y.o.) are estimated with a parametric global method, systematically varying the location of the division points until the total residual error attains a minimum. We used the sum of squared differences between the signal values and the predictions of the least-squares linear fit through the values (see details in Methods). The change points were the same for men and women (Fig. 1e). The frequency of short sleepers (those reporting sleeping on average 5h) increases with age, whereas the frequency of long sleepers (those reporting sleeping on average 9h or more) shows a U-shaped association with age (see Fig. 1d). Sex differences are overall larger among long sleepers. There were predominantly more women among long sleepers and mostly in the first two age groups. There were more men reporting shorter sleep among younger participants but this pattern reverses among older participants with more women reporting short sleep. A linear-mixed model (LMM) was calculated to predict reported sleep duration with age, age² (the quadratic component of age, since Fig. 1 shows a U-shaped curve), gender, education, home environment, commute duration and their interaction with age as fixed effects, and random intercepts clustered by countries: sleep duration ~ age*(gender + education + home environment + commute duration) + age² + (1 | country).

We included these variables as their association with sleep duration has been reported in the literature. All variables significantly predicted reported sleep duration, the results of this LMM are detailed in Table S2, and the evolution of reported sleep duration as a function of age by gender, education level, commute duration and home environment is displayed in Fig. 1e. Given the magnitude of the dataset, we chose to focus on effect sizes, which are independent of the sample size. Age is strongly associated with reported sleep duration (Hedge's $g = 0.49$, 95% CI = [0.47, 0.51]), so is age² (Hedge's $g = 0.50$, 95% CI = [0.48, 0.52]]) and gender (Hedge's $g = 0.12$, 95% CI = [0.12, 0.13]). The Hedge's $g$ of age is computed between the two extrema of reported sleep duration (maximum at 19 y.o., minimum at 50 y.o.). There is a strong interaction between the effect of age and education. Across all participants the effect size of education is weak (Hedge's $g = 0.059$, 95% CI = [0.054, 0.064]), but if we only consider participants below 22 y.o. (when participants in tertiary education would still be students) this effect rises to $g = 0.23$, 95% CI = [0.22, 0.24]), tertiary education leading to shorter declared sleep durations. Commute duration is also associated with reported sleep duration, with shorter self-reported sleep duration for participants with more than 1 h of commute compared to participants with 30 min to 1 h per day (Hedge's $g = 0.10$, 95% CI = [0.10, 0.11]). However, we do not find a linear relationship between amount of commute time and reported sleep duration; participants who commute less than 30 min report less sleep duration than those

commuting 30 min to 1 h (Hedge's $g = 0.06$, 95% CI = [0.055, 0.066]). Finally, we report a very weak association between home environment and reported sleep duration (Hedge's $g = 0.03$, 95% CI = [0.02, 0.03]).

### Reported sleep duration is associated with cognitive function in the third phase of adult life

To provide a reliable estimate of spatial navigation ability, we examined the data from a subset of participants who completed at least the first four wayfinding levels ($N = 418,152$)[18]. To quantify spatial ability, we defined the "wayfinding performance" metric (WF), which captures how efficient participants were in completing the wayfinding levels, while correcting for video-gaming skills. We also defined the "training performance" metric, which captures how efficient participants were in completing the tutorial levels, where navigation skills are not required (see Methods). To compare these two metrics, we normalized them (z-score) across all participants. Two ANOVAs were calculated to predict training and wayfinding performance with sleepduration, sleepduration² (the quadratic component of sleep duration, since Fig. 2b shows an inverted U-shaped curve) and their interaction with age groups, gender, education, home environment, commute duration as independent variables: performance ~ agegroups*(sleepduration + sleepduration²) + education + home environment + commute duration.

The 3 age groups are the ones identified in Fig. 1c: younger (19–33 y.o., $N = 191,791$), middle (34–53 y.o., $N = 153,884$), and older (54–70 y.o., $N = 72,477$) participants. Figure 2b shows the training and wayfinding performances as a function of the reported sleep duration, and Table S3 reports the results of both ANOVAs. The magnitude of both the linear and quadratic terms of reported sleep duration is larger for the wayfinding performance ($\beta_{sleepduration} = 0.39$, 95%CI = [0.33 0.46], $\beta_{sleepduration^2} = -0.028$, 95%CI = [−0.033 −0.024]) than for the training performance ($\beta_{sleepduration} = 0.14$, 95%CI = [0.07 0.22], $\beta_{sleepduration^2} = -0.012$, 95%CI = [−0.017 −0.010]). Furthermore, the interaction of both the linear and quadratic terms of reported sleep duration with the age groups are significant for wayfinding but not training performance (see Table S3). The optimal wayfinding performance was achieved by participants who reported sleeping 7 h in all age groups. However, the association between reported sleep duration and wayfinding performance was most significant for 54–70 years participants, and the inverted U-shaped pattern was stronger in this last age group, see Supplementary Fig 5. This pattern also held when stratifying participants by gender and level of education, see Supplementary Fig 6. Reported sleep duration is not uniformly globally distributed, with country-level average being almost 1 h more in some countries than in others (e.g., Japan: $M = 6.63$ h, SD = 0.04 h, Albania: $M = 7.54$ h, SD = 0.04 h, Hedge's $g = 0.79$, 95%CI = [0.68 0.89]), see Fig. 3b, c. To control for the other demographics (age, gender, education, commute duration and home environment), we computed the country-level conditional modes from the above LMM, i.e., the difference between the population-level average predicted reported sleep duration given our fixed effects and the reported sleep duration predicted for each country, see Fig. 3b. The correlation between countries' conditional mode and their raw average self-reported sleep duration was very strong ($r = 0.97$, $p < 0.001$). Notably, the inhomogeneities in reported sleep duration do not seem to be random, but can be clustered into supra-national regions. We used the 7 global clusters of countries defined in ref. 23, see Fig. 3a and Fig. S3. These clusters are based on historical and economical proximity and constitute one of the most common world map segmentation[24]. To test the significance of this global clustering, we computed an ANOVA predicting the reported sleep durations from the age, gender, and global clusters. We obtained a significant effect of all predictors: global clusters ($F(6, 730181) = 936.97$, $p < 0.001$), gender ($F(1, 730181) = 4001.28$, $p < 0.001$), and age ($F(1, 730181) = 7576.27$, $p < 0.001$), and the significance remained for the three phases of adult life. To further assess

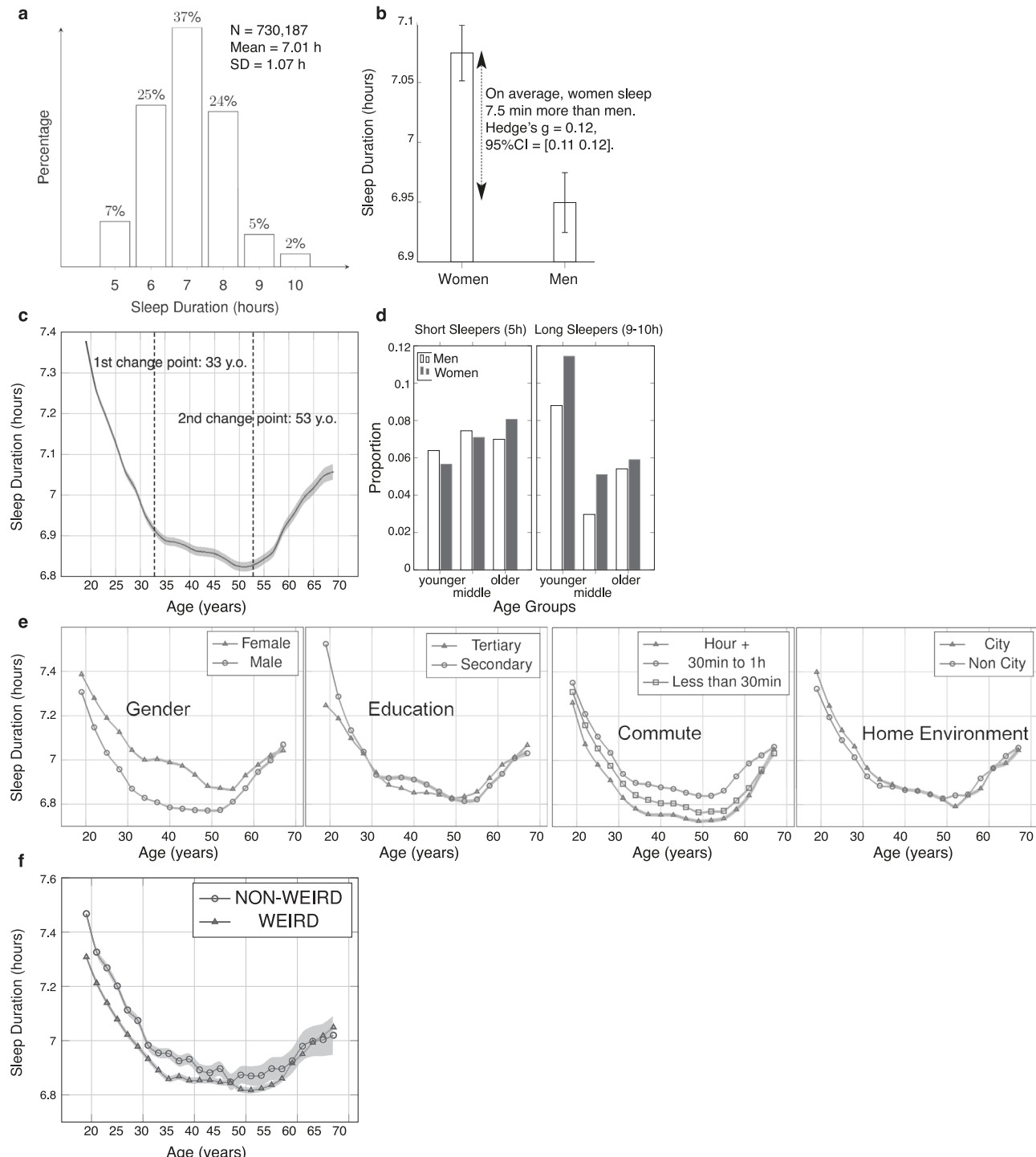

**Fig. 1 | Distribution of reported sleep duration. a** Distribution across 730,187 participants over 63 countries ($M = 7.01$ h, SD = 1.07 h). **b** Reported sleep duration is shorter for men ($M = 6.94$ h, SD = 1.04 h) than women ($M = 7.07$ h, SD = 1.09 h). **c** Reported sleep duration across the adult lifespan. This evolution can be split in 3 phases: a sharp decrease from 19 years to 33 years, a slower decrease from 34 years to 53 years, and a re-increase from 54 years onwards. **d** Proportion of short (5 h) and long (9–10 h) sleepers per gender and age group. **e** Reported sleep duration across age for each gender (381,153 men, 349,034 women), level of education (204,017 secondary, 526,170 tertiary), commute duration (291,822 less than 30 min, 254,362 30 min to 1 h, 183,764 h plus) and home environment (222097 city, 508090 non city). **f** Reported sleep duration across age for WEIRD ($n = 526,136$) and non-WEIRD ($n = 204,051$) populations. WEIRD stands for Western, Educated, Industrialized, Rich and Democratic[26]. Data points correspond to the average reported sleep duration within 3-year windows. Error bars correspond to 95% confidence intervals.

the significance of the global clustering, we randomly shuffled the country labels 100 times. At each iteration, we computed the $F$-value from an ANOVA predicting the reported sleep durations from the age, gender, and global clusters based on the random country labels. Fig. S1 shows the histogram of the bootstrapped random global clusters $F$-values ($M = 0.95$, 95%CI = [0.84 1.06]) and the true $F$-value corresponding to the actual global clusters (vertical red line, $F(6, 730181) = 936.97$), far above the largest random $F$-value. We

**a**   SHQ navigation   Training: level 1   Wayfinding: level 11

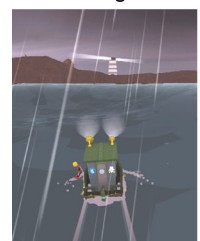 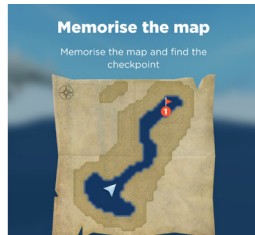 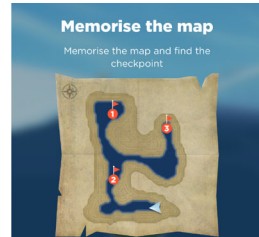

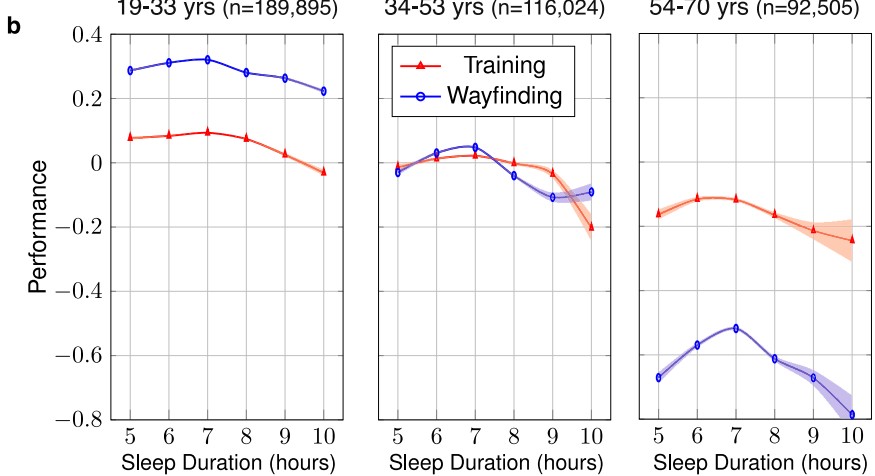

**Fig. 2 | Association between reported sleep duration and spatial ability in the 3 identified age groups. a** Wayfinding task: spatial ability was quantified with the Sea Hero Quest (SHQ) app. Participants completed 2 training levels that did not require spatial ability, as the target was visible from the starting point. They also completed 4 wayfinding levels, where the participants were asked to memorize a map, and navigate as quickly as possible toward 3 checkpoints in a set order. **b** Training (motor ability) and Wayfinding (spatial ability) performance averaged for each sleep duration. See Table S3 for statistical analysis. Error bars correspond to 95% confidence intervals.

repeated this analysis with a slightly different set of clusters, based on cultural similarities between countries[25], see Figs. S1, S3, and Methods. There is a positive correlation between the country-level average reported sleep duration and their distance to the equator, as shown in Fig. 3d ($r = 0.52$, $p < 0.001$). This association survives when controlling for GDP per capita (Fig. S2): a linear regression with reported sleep duration as the response variable gives significant effects for both GDP (Gross Domestic Product) per capita ($t(60) = -4.57$, $p < 0.001$) and latitude ($t(60) = 6.59$, $p < 0.001$). We computed the same regression with the aforementioned countries' conditional modes (country-level deviation from the population-level average predicted reported sleep duration corrected for the other demographics) as predictors, and also found a significant effect of both GDP per capita ($t(60) = -2.37$, $p = 0.02$) and latitude ($t(60) = 3.38$, $p = 0.001$). Henrich and colleagues showed that people from WEIRD (Western, Educated, Industrialized, Rich and Democratic) countries represent as much as 80% of participants in behavioral science studies, but only 12% of the world population[26]. To test whether our findings generalize to the populations less represented in the literature, we compared the average reported sleep duration between WEIRD (19 countries, $N = 545,639$) and non-WEIRD countries (44 countries, $N = 206,130$), labeled in Table S1. WEIRD and non-WEIRD reported sleep durations followed the same U-shaped pattern across age, with change points at 33 years and 55 years for the WEIRD population and 31 and 52 years for the non-WEIRD population, see Fig. 1f. We found no difference between the reported sleep durations of WEIRD ($M = 6.92$h, SD = 1.03) and non-WEIRD ($M = 7.12$ h, SD = 1.15) countries, even when controlling for GDP per capita and latitude: a linear regression with sleep duration as the response variable still gives significant effects for both GDP per capita

($t(59) = -3.43$, $p = 0.001$) and latitude ($t(59) = 3.13$, $p = 0.003$), but not for WEIRD status ($t(59) = 1.19$, $p = 0.24$).

## Discussion

To our knowledge this is the largest single study on self-reported sleep duration across the life-course and its modulation by gender, geographic location and economy. Although previous studies have also shown a non-linear association of sleep duration with age[9,16,27], here we reveal three distinct phases in the life-course where reported sleep duration changes at an approximately monotonic rate during each period. Such reductions in sleep to mid-life have previously been related to the demands of child-care and working life[28]. The increasing sleep reported after 53 years is likely related to a reduction in child-rearing responsibilities and alleviation of other factors driving the lower sleep in mid-life[29,30]. Notably, the specific change points we observed in the overall population were exactly matched across and across men and women, and very similar in the non-WEIRD populations (31 and 52 years). Thus, this pattern of reported sleep duration as a function of age appears robust across various populations.

Prior studies have highlighted the importance of sleep for memory performance[31]. However, few studies have examined the link between self-reported sleep duration and memory performance on a task across many countries and across the life-course. Here, we show a very limited relationship between self-reported sleep duration and performance in the first two phases of life-course, but a clear association in the last phase after 53 years of age, where reporting sleeping 7 h was linked to optimal performance. Our findings extend prior reports in small samples in WEIRD nations of an inverted U-shaped association of reported sleep duration with cognitive

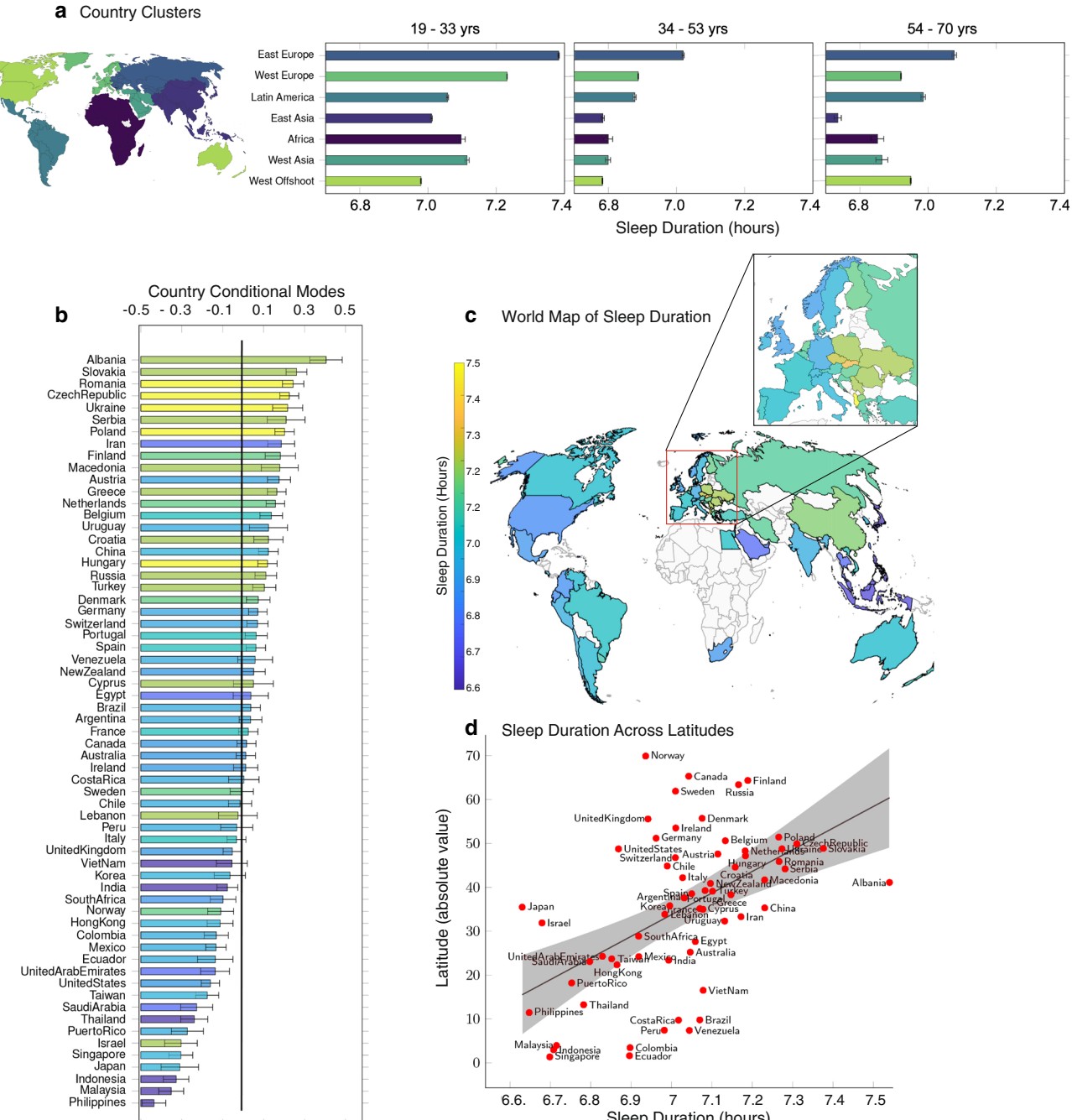

**Fig. 3 | Distribution of reported sleep duration across countries. a** Reported sleep durations averaged in 7 global clusters of countries defined by Maddison[23]. In the first, second and third age groups, the sample sizes are, respectively, $n = 321,406$, $n = 263,932$ and $n = 144,849$. The samples sizes of the country clusters are available in Supplementary Fig 3. **b** Random intercepts clustered by countries from a linear-mixed model predicting reported sleep duration with age, age[2], gender, education, home environment, commute duration and their interaction with age as fixed effects and country as random effect. Colors reflect raw reported sleep duration (not corrected for fixed effects), as in panel **c**. **c** World map of reported sleep durations. **d** For each country, average reported sleep duration as a function of the absolute value of its average latitude (Pearson's $r = 0.52$, $p < 0.001$). Error bars correspond to the standard errors.

performance[1,32–34] and cognitive decline in older people[35,36] to assessment across 63 countries, including both WEIRD and Non-WEIRD nations. Interestingly, we found this U-shape in spatial navigation performance, but not in motor skills (training performance). This is consistent with the fact that short and long sleepers are not impaired evenly across all cognitive domains[37,38]. Not obtaining enough sleep (<6h) would be expected to impact memory performance based on multiple neurophysiological mechanisms underlying the cognitive functions of sleep. The causal link between longer sleep duration and poorer cognition, however, is not yet clarified. Why would reporting

sleeping longer lead to poorer memory? A plausible explanation is that long sleep could be a marker of an underlying medical condition associated with cognitive decline, such as depression[39]. In addition, sleeping more than 9 h a night may be linked to more fatigue and disrupted sleep[40].

We found that the pattern of reported sleep duration across age was remarkably stable across countries. On the contrary, the average country-level reported sleep duration was highly geographically clustered. We explore what mechanisms might underlie this. Two factors appear to mediate this: cultural norms and latitude. How sleep is

considered substantially vary across countries. For instance workplace napping can be frowned upon in some countries (e.g., the US or France) while considered normal by their neighbors (e.g., Spain or Italy)[41]. Far East countries, Japan in particular, have been noted to report less sleep[10,11]. This could be partly explained by the work culture in these countries, Japan being the only nation in the world in which 25% of employed workers work more than 50 h per week[12]. Here, we extend this to 63 countries and find strong clustering in the data, consistent with geographic norms spreading across local regions. Another factor suspected of driving sleep duration across countries has been latitude, as suggested by sleep durations recorded from South to North Chile[42]. Here, we reveal this pattern across 63 nations. Prior research has found that a country's average bedtime, but not average wake time, predicts sleep duration, and that solar cues do influence sleep but are being attenuated in the real world—compared to lab-based experiments, particularly around bedtime[43]. This likely increases the importance of cultural and socioeconomic factors relative to latitude in mediating the effects of geographical location on sleep duration.

In this study, we collected and analyzed self-reported sleep durations. Self-reported sleep durations may differ from actual sleep patterns and the concordance may be modulated by age, sex[44], and sleep disorders[45]. Since participant demographics and sleep disorders are also associated with spatial ability[19,46], it will be important to validate our results with objective sleep parameters recorded using polysomnography. We controlled for the effect of age, gender, level of education, commute duration, home environment and country on sleep duration. Other participant-related variables such as ethnicity, socioeconomic status, neurological and psychiatric disorders, and a high level of inflammation also have been associated with sleep duration variation[5,7,47,48]. External variables such as the day of the week, the time of the day, and light levels when the cognitive test was performed significantly impact sleep duration and cognitive performance[10,49,50]. We unfortunately did not collect these variables in our current dataset, but they would be important to include in a future model. Future studies should also assess sleep quality known to be confounded with sleep duration.

Our data reveal a set of patterns linked to sleep across the world population. We show two major inflection points in the trajectory of reported sleep duration throughout the human life course and find that geographical and cultural factors can be used to predict the average reported sleep of a nation. We also show that objective memory performance can be associated with self-reported sleep duration, 7 h representing a rather universal optimum self-reported sleep duration for cognition in adults. Importantly, declared sleep duration appears to be associated with memory performance starting with late mid-life.

Crossing the geographical, linguistic, and cultural boundaries to generalize empirical evidence on sleep has been curbed by methodological difficulties. Collecting a large amount of sleep-related data in a standardized and reliable manner from multiple geographical regions across the world remains a challenge. The joint collection of sleep data with behavioral data is even more challenging, albeit essential to understand the associations between sleep, cognition, and behavior. The current study shows that the spread of video games across the planet can be harnessed by sleep researchers to access larger and more diverse populations, allowing them to replicate findings and make discoveries more likely to be valid at a global scale.

## Methods

### Informed consent and ethics approval

This study has been approved by the UCL Ethics Research Committee. The ethics project ID number is CPB/2013/015. Participants were explained the purpose of the game when opening the app, before they could start playing. Players had to tick an 'opt in' box if they agreed to share their data with us. They were guided to the settings where the opt out option always remained available. They could also choose to provide their demographic or not. This was done in two steps. First, they could enter their age, gender and home country. Then, after having played a few levels, participants could provide further information such as their average sleep duration, level of education, commute duration, and the type of environment they grew up in.

### Data and task

The Sea Hero Quest app was released on May 2016 on the App Store for iOS and on Google Play for Androids. It was available to the general public in 17 languages: English, French, German, Spanish, Macedonian, Greek, Croatian, Dutch, Albanian, Hungarian, Romanian, Slovak, Czech, Polish, Portuguese, Italian, and Serbian. The Sea Hero Quest task and data collection have been thoroughly described in ref. 18. We set up a portal where researchers can invite a targeted group of participants to play Sea Hero Quest and generate data about their spatial navigation capabilities. Those invited to play the game will be sent a unique participant key, generated by the Sea Hero Quest system according to the criteria and requirements of a specific project. https://seaheroquest.alzheimersresearchuk.org/Access to the portal will be granted for non commercial purposes. Future publications based on this dataset should add "Sea Hero Quest Project" as co-author.

**Participants.** A total of 3,881,449 participants played at least one level of the game. 60.8% of the participants provided basic demographics (age, gender, home country) and 27.6% provided more detailed demographics (home environment, level of education). We removed participants above 70 years old because we have previously shown that they suffer from a strong selection bias, causing their performance to be substantially higher than would be expected in unselected participants of the same age[18]. We removed participants with an average sleep duration above 10 h (0.6%) or below 5 h (0.9%) for 2 reasons. First, they represent only 1.5% of the participants, much less than the other sleep duration groups. Second, they are more likely than others to erroneously enter their demographics. As explained in ref. 18, there are many more 99ers than would be predicted from the age distribution, likely because it is one extremum of the age range. We noticed that the proportion of 99ers in participants who reported sleeping extreme hours is much higher than in participants who reported sleeping between 5 h and 10 h. Thus, we applied the same treatment to 99ers as to participants who reported sleeping extreme hours and removed them from the analysis. We compared the wayfinding performance of participants who chose to report sleep duration versus those who didn't. We obtained a very weak effect size: Hedge's $g = 0.012$, 95% CI = [0.008, 0.016], indicating that there is no meaningful difference in terms of navigation between the two groups. To ensure robust results within countries, we removed participants from countries with fewer than 500 players. This resulted in 730,187 participants from 63 countries included in our analysis, (see Table S1). To provide a reliable estimate of spatial navigation ability, we examined the data from a subset of participants who completed the tutorial levels at least the first four wayfinding levels (up to level 11 in the game, $N = 418,152$). We only analyzed the first attempt of each level. When the participant retried a level to improve their score, we did not include the associated trajectory in the analysis.

**Demographics.** Among the 730,187 included participants, 381,153 identified as men and 349,034 as women. The mean age was 38.71 years (SD = 14.53 years). 526,170 reported having received a tertiary education and 204,017 a secondary education or less. 222,097 reported having grown up in a city environment and 508,090 outside cities. 291,822 reported commuting on average less than 30 min per day, 254,362 between 30 min and 1 h, and 183,764 more than 1 h. Among the

418,152 participants included for the spatial ability analysis, 224,159 identified as men and 193,993 as women. The mean age was 37.80 years (SD = 14.13 years). 301,515 reported having received a tertiary education and 116,637 a secondary education or less. 123,733 reported having grown up in a city environment and 294,419 outside cities. 162,680 reported commuting on average less than 30 min per day, 148,167 between 30 min and 1 h, and 107,305 more than 1 h.

**Country clusters.** We used the supra-national clusters defined in ref. 23. The sample size and demographics of each cluster are available in Supplementary Fig 3.

- African cluster: South Africa, Egypt.
- East Europe cluster: Czech Republic, Hungary, Poland, Romania, Russia, Slovakia, Ukraine, Albania, Serbia, Croatia, Albania, Macedonia, Cyprus.
- Western Europe cluster: Germany, Austria, Switzerland, England, Ireland, Sweden, Norway, Finland, Denmark, Netherlands, France, Belgium, Italy, Portugal, Spain, Greece.
- Western Asia cluster: United Arab Emirates, Saudi Arabia, Turkey, Israel, Lebanon, Iran.
- Western Offshoot: United States, Canada, Australia, New Zealand.
- Latin America cluster: Argentina, Peru, Chile, Bolivia, Colombia, Puerto Rico, Ecuador, Costa Rica, Venezuela, Uruguay, Brasil, Mexico.
- East Asia cluster: India, Indonesia, Malaysia, Philippines, Thailand, Vietnam, China, Singapore, Hong Kong, Taiwan, Japan, South Korea.

We also used the cultural clusters defined in ref. 25. The sample size and demographics of each cluster are available in Supplementary Fig 4.

- African cluster: South Africa, Egypt.
- East Europe cluster: Czech Republic, Hungary, Poland, Romania, Russia, Slovakia, Ukraine, Albania, Serbia, Croatia, Albania, Macedonia, Cyprus.
- Near East cluster: Greece, Turkey, Lebanon, Iran.
- Germanic cluster: Germany, Austria, Switzerland.
- Nordic cluster: Sweden, Norway, Finland, Denmark, Netherlands.
- Latin Europe cluster: France, Belgium, Italy, Portugal, Spain, Israel.
- Anglo cluster: United States, Canada, Australia, New Zealand, England, Ireland.
- Arabic cluster: Saudi Arabia, United Arab Emirates
- Latin America cluster: Argentina, Peru, Chile, Bolivia, Colombia, Puerto Rico, Ecuador, Costa Rica, Venezuela, Uruguay, Brasil, Mexico.
- Confucian Asia cluster: China, Singapore, Hong Kong, Taiwan, Japan, South Korea.
- Far East cluster: India, Indonesia, Malaysia, Philippines, Thailand, Vietnam

## Spatial navigation task and metric

**Navigation task.** At the beginning of each wayfinding level, participants were asked to memorize the locations of 1 to 5 checkpoints to visit on a map. The map disappeared, and they had to navigate a boat through a virtual environment to find the different checkpoints in a set order (Fig. 2a). Participants were incentivized to complete the task as quickly as possible; they were awarded 'stars' when finishing each level before a set time.

**Wayfinding and training performances.** We collected the coordinates of participants' trajectories every 500 ms. As in ref. 18, we computed the trajectory length in pixels, defined as the sum of the Euclidean distance between the points of the trajectory. To control for familiarity with handling tablets/smartphones, we divided the trajectory length of each level by the sum of the trajectory lengths of the first two levels. The first two levels were tutorial levels to familiarize the participant with the game commands. They did not require any spatial ability as the target was visible from the starting point. We defined the wayfinding performance (WF) as the 1st component of a principal component analysis across the trajectory lengths (divided by the tutorial trajectory lengths) of the first four wayfinding levels (levels 6, 7, 8 and 11, 60.14% of variance explained). This metric being based on the trajectory length, it varies as the opposite of the performance: the longer the trajectory length, the worse the performance. We took the additive inverse of the metric and added an offset, so that WF = 0 corresponds to the worst performances. The training performance (TP) was obtained via the same method, but only with the first two tutorial levels, and obviously without controlling for familiarity with handling tablets/smartphones. Finally, to put TP and WF on the same scale, we z-scored them ($M = 0$, SD = 1). To sum up, WF and TP vary in the opposite direction than the trajectory length. Short, thus optimal trajectories, correspond to a high performance. Positive WF and TP values correspond to above average performance, while negative values correspond to below average performance.

## Analysis

**Change-points estimation.** The change points of the curve representing reported sleep duration as a function of age (Fig. 1c) are estimated with a parametric global method, systematically varying the location of the division points until the total residual error attains a minimum[51]. This method is implemented in the Matlab function (introduced in R2016a) *findchangepts* with the linear statistics option. More precisely, this method starts by randomly dividing the age curve into two sections. Then it computes a least-square regression line for each section, and adds the deviations (sum of squared differences between the signal values and the predictions) section-to-section to find the total residual error. This procedure is repeated for all divisions points until the total residual error attains a minimum. Since the number of change points is to be determined, we added a penalty term to the residual error. This is necessary since the residual error decreases with the number of change points and too many change points results in overfitting. We used a penalty term that grows linearly with the number of change points. Let $x$ be the signal and $\hat{x}$ be the best-fit line between $s_j$ and $s_{j+1}$, thus the sum of square error is

$$\text{SSE}_j = \sum_{i=s_j}^{s_{j+1}-1} (x_i - \hat{x}_i)^2 \qquad (1)$$

If there are $K$ changepoints to be found, with $s_0$ the first and $s_K$ the last sample of the signal, we minimized

$$\text{Deviation}_K = \sum_{j=0}^{K-1} \text{SSE}_j + \beta K \qquad (2)$$

The constant $\beta$ corresponds to a fixed penalty for each added change point. The algorithm does not add additional change points if the decrease in deviation does not meet a set threshold $T$. We incrementally varied the threshold from $T = 0.005$ to $T = 0.09$ by 0.001 steps and obtained the same change points (at 33 years and 53 years). At $T = 0.1$, the algorithm returns a unique change point at 38 years. At $T = 0.004$, the algorithm returns 3 change points: at 21 years, 33 years, and 53 years. Since the curve representing sleep duration as a function of age does not show any inflection point at 38 years or 21 years are not, we set the threshold to $T = 0.02$.

**Linear-mixed models.** The parameters of the linear-mixed models have been estimated with the maximum likelihood method, and the

covariance matrix of the random effects have been estimated with the Cholesky parameterization.

### Reporting summary

Further information on research design is available in the Nature Portfolio Reporting Summary linked to this article.

## Data availability

The data necessary to reproduce the results presented in this manuscript is available at https://osf.io/j7sd4/?view_only=92c1aa56abfc4d0f99501c9f260fa324

## Code availability

The code (Matlab R2018a) necessary to reproduce the results presented in this manuscript is available at https://osf.io/j7sd4/?view_only=92c1aa56abfc4d0f99501c9f260fa324

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

## Acknowledgements

The authors wish to thank Deutsche Telekom for supporting and funding this research and Alzheimer Research UK (ARUK-DT2016-1-HS MH) for funding the analysis, the Glitchers Limited for the design and game production.

## Author contributions

H.S., M.H., and A.C. supervised the project, H.S., M.H., A.C., E.M., R.D., J.W., and C.H. designed research; A.C. analyzed the data; A.C., A.L., M.R., and H.S. wrote the paper.

## Competing interests

The authors declare no competing interests.
