## [Peer Review File · Nature Communications]

Reported sleep duration segments the adult life-course into three phasesREVIEWER COMMENTS

Reviewer #1 (Remarks to the Author):

This is a very interesting article, I commend the authors for their work. My main comment after reviewing the manuscript relates to the causal language used throughout the manuscript. The authors frame the discussion as if self reported sleep duration is per se causing the cognitive impairment, rather than being a proxy measure or barometer of their overall sleep health. In page 9, the authors did a good job at describing this issue, but I would prefer if they could be more precise about what they really measured: For example, "Why would increased sleep lead to poorer memory", perhaps better to say "why reporting sleeping longer...". In general, I would recommend toning down the causal language, and rather than stating that sleeping 7h as the optimal amount, being more honest about what was really measured and using as much as possible in the discussion something along the lines of "optimal self-reported sleep duration.." (instead of "a rather universal optimum sleep duration for cognition in adults"). [you mostly did this, with some exceptions]

When defining the acronym WEIRD, the W stands for Western and rich is typed twice. Why did you choose this category? It comes out of nowhere. Lumping together western countries (such as Latin America) and Asian countries seems baseless.

Do you have any measure of rural vs urban environment?

Another reason for sleep duration differences may be related to stigma, as in some societies longer sleep is viewed badly.

Please add a paragraph about the limitations of your study, and include a statement about the generalizability of your sample.

Reviewer #2 (Remarks to the Author):

This is a very unique paper examining reported self-reported sleep duration in a large multi-national sample across the adult life-span. Using strong statistical modeling, the paper finds three consistent age periods related to sleep duration across cultures, including a young adult period, transition at 34 and another transition after 53 years. This study also uniquely utilizes a well-validated remote navigational experiment to show a u-shaped relationship with sleep duration. The findings are important and extremely novel, but I have several methodological considerations that should be discussed in a limitations section or further analysis.

Major points:

1. Self-reported sleep duration has dubious correlations with objectively measured sleep across many studies (Trimmel et al., 2021). The authors should discuss this limitation and its impact on their results.
2. The authors need to convince the reader that this study does not suffer from selection bias. First, what is the precedent for excluding sleep <5 and >10 hours? Next, are there any major differences in wayfinding between those who chose to report sleep duration vs. not?
3. Do the authors have any data on minority status? They have included home environment (not defined—see below), but race/ethnicity is an extremely important variable to include when understanding sleep duration variation. Finally, while the inclusion of such a large sample across countries is extremely unique and valuable, a large portion of the sample was highly educated, suggesting a limitation to the broad applicability of the results. Regardless, anything in the data that the authors could examine to speak to this issue would strengthen the findings.

Minor:

- Figures:
 - o pg. 3/16: Figure 1B:
 - ♣ caption units should be reported in hours, not years.
 - ♣ WEIRD is used in caption/figure but has not yet been defined.
 - o Fig 3A: part of alaska is cut off on the country clusters map
 - WEIRD- Please check definition of - pg 8 ("WEIRD stands for Rich, Educated, Industrialized, Rich and Democratic ") and actually this should be defined much earlier in paper, before it's used in Figure 1)
 - p 8/16: I'm not sure this sentence is accurate: "Notably, the specific change points we observed in the overall population were exactly matched across WEIRD and non-WEIRD populations, and across men and women." because on p. 7 the authors write: "change points at 33 yrs and 55 yrs for the WEIRD population and 31 and 52 yrs for the non-WEIRD population, see Fig 1F" maybe they weren't statistically different?
 - Methods: How is home environment measured? Please define.
 - Some more information could be included about how the participants accessed the program. I know it is available on apple devices, but more information should be included here (it matters as some countries may have differences in accessibility to the program, and this should be discussed).

Trimmel, K., Eder, H. G., Böck, M., Stefanic-Kejik, A., Klösch, G., & Seidel, S. (2021). The (mis) perception of sleep: factors influencing the discrepancy between self-reported and objective sleep parameters. *Journal of Clinical Sleep Medicine*, 17(5), 917-924.

Reviewer #3 (Remarks to the Author):

In the interesting study by Coutrot and colleagues, the authors analyzed subjective sleep duration from a large cohort of participants across 63 countries and they identified three distinct phases in the adult human life course; within the late adulthood (> 54 yrs to 70 yrs) the authors identified an inverted U-shape regulation between sleep duration and cognitive performance as measured by spatial navigation with a peak of optimal performance observed at 7 hours of sleep.

This study has definitively major strengths including the large samples size and the generalizability of the findings across several countries in the world. In addition, the merit of the study is that it highlights the negative effects of long sleep duration on cognitive performance in a specific age group (late adulthood) which can address future research to better understand within this age group at higher risk for cognitive impairment, potential mechanisms for the observed association. The authors might want to highlight in the discussion how their findings can impact future research in the field of sleep and cognition.

These findings are in line with the growing body of epidemiological studies showing that prolonged sleep is statistically associated with significant morbidity as well as mortality. In many studies, this association was even stronger than that found with short sleep.

As discussed by the authors, a proposed hypothesis is that depression (either as a confounder or as a causal intermediate) may explain the statistical association between long sleep and poor health. Comorbid sleep disorders, low socioeconomic status, preclinical medical disease, and a high level of inflammation have also been proposed as potential explanations for the association between long sleep and elevated mortality risk.

Unfortunately, with the exception of the economic status, the other proposed mediators have not been taken into account by this study and the authors might want to address them as limitations of their study.

Among the factors that could have significantly affected the findings, there are also the time of the

day when the cognitive test was performed (circadian component) and light levels. In fact, both these factors can significantly impact alertness and cognitive performance. The authors should address these as possible limitations in the discussion.

There are aspects of the manuscript that might be improved to favor clarity and readability:

- Can the authors explain what age² and education² stand for?
- Can the author simplify the definition of what wayfinding performance represents?
- In the method section, It is not clear how many times the navigation task was performed by participants, please clarify.
- The authors state that they remove participants > 70 yrs old as they have a strong selection bias: can the authors better explain the nature of the selection bias?
- Looking at Figure 2, a U shape pattern is also detectable for young and middle-aged adults; did the author check for a linear trend across age groups? is it possible that the aging process and the inter-individual variability associated with aging in many physiological processes (sleep and cognition included) makes the U pattern more evident in older adults. Please, expand on this.

Point-by-point response to the reviewers' comments

Manuscript number: NCOMMS-22-10397

Author names: Coutrot A., Lazar A. S., Richards M., Manley E., Wiener J. M., Dalton R. C., Hornberger M. and Spiers H. J.

Title: Reported sleep duration segments the adult life-course into three phases

We thank all reviewers for their helpful assessment of our manuscript.

We quote Reviewers and Editor in Black font, our response in Blue and our revised added text to the manuscript in Red font with quotes.

REVIEWER COMMENTS

Reviewer #1 (Remarks to the Author):

This is a very interesting article, I commend the authors for their work. My main comment after reviewing the manuscript relates to the causal language used throughout the manuscript. The authors frame the discussion as if self reported sleep duration is per se causing the cognitive impairment, rather than being a proxy measure or barometer of their overall sleep health. In page 9, the authors did a good job at describing this issue, but I would prefer if they could be more precise about what they really measured: For example, "Why would increased sleep lead to poorer memory", perhaps better to say "why reporting sleeping longer...". In general, I would recommend toning down the causal language, and rather than stating that sleeping 7h as the optimal amount, being more honest about what was really measured and using as much as possible in the discussion something along the lines of "optimal self-reported sleep duration.." (instead of "a rather universal optimum sleep duration for cognition in adults"). [you mostly did this, with some exceptions]

We removed the causal language and specified that we used reported sleep durations. In particular, we modified the following sentences:

Abstract (page 1):

"During the third phase, where subjective sleep duration increases with age, cognitive performance, as measured by spatial navigation, was found to have an inverted u-shape relationship with reported sleep duration: optimal performance peaks at 7 hours reported sleep. World-wide subjective sleep duration patterns are geographically clustered, with declared sleep duration associated with economy, culture, and latitude. Eastern Europeans reported sleeping the most and Far East populations the least. These results reveal novel patterns in a core metric of health over the adult life-course and across the planet."

Page 5:

"Reported sleep duration is associated with cognitive function in the third phase of adult life"

Page 9:

"We also show that objective memory performance can be associated with subjective sleep duration, 7 hours representing a rather universal optimum self-reported sleep duration for cognition in adults. Importantly, declared sleep duration appears to be associated with memory performance starting with late midlife."

When defining the acronym WEIRD, the W stands for Western and rich is typed twice. Why did you choose this category? It comes out of nowhere. Lumping together western countries (such as Latin America) and Asian countries seems baseless.

We corrected our mistake in the acronym definition, thanks for spotting.

This category is defined in [23], where Henrich et al. notice that people from WEIRD countries represent as much as 80 percent of study participants, but only 12 percent of the world population. Henrich et al. argue that the results obtained from this very specific population do not necessarily generalize to the human species, and that researchers should seek more representative study participants.

Here we used the WEIRD category to show that our results are also valid in populations that are less present in the sleep research literature. We agree with the reviewer that this country category does not have a cultural or economical coherence, this is why we also used the clusters defined by Maddison [24] and Ronen & Shenkar [25].

[24] Maddison, A. *The world economy* (OECD publishing, 2006).

[25] Ronen, S. & Shenkar, O. Mapping world cultures: Cluster formation, sources and implications. *Journal of International Business Studies* 44 (9), 867–897 (2013) .

[23] Henrich, J., Heine, S. J. & Norenzayan, A. The weirdest people in the world? *Behavioral and Brain Sciences* 33 (2-3), 61–135 (2010)

We added the following paragraph (page 8):

“Henrich and colleagues showed that people from WEIRD (Western, Educated, Industrialized, Rich and Democratic) countries represent as much as 80% of participants in behavioral science studies, but only 12% of the world population [23]. To test whether our findings generalize to the populations less represented in the literature, we compared the average reported sleep duration between WEIRD (19 countries, N=545,639) and non-WEIRD countries (44 countries, N=206,130), labeled in Table S1.”

Do you have any measure of rural vs urban environment?

As for the other demographics, the home environment is self-reported.

We added a citation to our recent paper specifically dedicated to the association between home environment and spatial ability:

[20] Coutrot, A. et al. Entropy of city street networks linked to future spatial navigation ability. *Nature* 604 (7904), 104–110 (2022) .

Another reason for sleep duration differences may be related to stigma, as in some societies longer sleep is viewed badly.

We added the following paragraph (page 9):

“How sleep is considered substantially vary across countries. For instance workplace napping can be frowned upon in some countries (e.g. the US or France) while considered normal by their neighbors (e.g. Spain or Italy) [40].”

[40] Alger, S. E., Brager, A. J. & Capaldi, V. F. Challenging the stigma of workplace napping. *Sleep* 42 (8) (2019) .

Please add a paragraph about the limitations of your study, and include a statement about the generalizability of your sample.

We added the following paragraph (page 9):

“In this study, we collected and analyzed self-reported sleep durations. Self-reported sleep durations may differ from actual sleep patterns and the concordance may be modulated by age, sex [43], and sleep disorders [44]. Since participant demographics and sleep disorders are also associated with spatial ability [19, 45], it will be important to validate our results with objective sleep parameters recorded using polysomnography. We controlled for the effect of age, gender, level of education, commute duration, home environment and country on sleep duration. Other participant-related variables such as ethnicity, socioeconomic status, neurological and psychiatric disorders, and a high level of inflammation also have been associated with sleep duration variation [5, 7, 46, 47]. External variables such as the day of the week, the time of the day, and light levels when the cognitive test was performed significantly impact sleep duration and cognitive performance [10, 48, 49]. We unfortunately did not collect these variables in our current dataset, but they would be important to include in a future model. Future studies should also assess sleep quality known to be confounded with sleep duration.”

Reviewer #2 (Remarks to the Author):

This is a very unique paper examining reported self-reported sleep duration in a large multi-national sample across

the adult life-span. Using strong statistical modeling, the paper finds three consistent age periods related to sleep duration across cultures, including a young adult period, transition at 34 and another transition after 53 years. This study also uniquely utilizes a well-validated remote navigational experiment to show a u-shaped relationship with sleep duration. The findings are important and extremely novel, but I have several methodological considerations that should be discussed in a limitations section or further analysis.

Major points:

1. Self-reported sleep duration has dubious correlations with objectively measured sleep across many studies (Trimmel et al., 2021). The authors should discuss this limitation and its impact on their results.

We added the following paragraph (page 9):

“In this study, we collected and analyzed self-reported sleep durations. Self-reported sleep durations may differ from actual sleep patterns and the concordance may be modulated by age, sex [43], and sleep disorders [44]. Since participant demographics and sleep disorders are also associated with spatial ability [19, 45], it will be important to validate our results with objective sleep parameters recorded using polysomnography. We controlled for the effect of age, gender, level of education, commute duration, home environment and country on sleep duration. Other participant-related variables such as ethnicity, socioeconomic status, neurological and psychiatric disorders, and a high level of inflammation also have been associated with sleep duration variation [5, 7, 46, 47]. External variables such as the day of the week, the time of the day, and light levels when the cognitive test was performed also significantly impact sleep duration and cognitive performance [10, 48, 49]. We unfortunately did not collect these variables in our current dataset, but they would be important to include in a future model. Future studies should also assess sleep quality known to be confounded with sleep duration.”

2. The authors need to convince the reader that this study does not suffer from selection bias. First, what is the precedent for excluding sleep <5 and >10 hours? Next, are there any major differences in wayfinding between those who chose to report sleep duration vs. not?

We added the following supplemental figure (Fig S5) and paragraph (page 10):

“We removed participants with an average sleep duration above 10h (0.6%) or below 5h (0.9%) for 2 reasons. First, they represent only 1.5% of the participants, much less than the other sleep duration groups. Second, they are more likely than others to erroneously enter their demographics. As explained in [18], there are many more 99ers than would be predicted from the age distribution, likely because it is one extremum of the age range. We noticed that the proportion of 99ers in participants who reported sleeping extreme hours is much higher than in participants who reported sleeping between 5h and 10h. Thus we applied the same treatment to 99ers as to participants who reported sleeping extreme hours and removed them from the analysis. We compared the wayfinding performance of participants who chose to report sleep duration versus those who didn't. We obtained a very weak effect size: Hedge's $g = 0.012$, 95% CI = [0.008, 0.016], indicating that there is no meaningful difference in terms of navigation between the two groups.”

Fig S4 - Age distribution of participants who reported sleeping less than 5h, between 5h and 10h, more than 10h.

3. Do the authors have any data on minority status? They have included home environment (not defined—see below), but race/ethnicity is an extremely important variable to include when understanding sleep duration variation.

See the new limitation paragraph in response to comment 1.

Finally, while the inclusion of such a large sample across countries is extremely unique and valuable, a large portion of the sample was highly educated, suggesting a limitation to the broad applicability of the results. Regardless, anything in the data that the authors could examine to speak to this issue would strengthen the findings.

Among the 418,152 participants included for the spatial ability analysis, 301,515 reported having received a tertiary education and 116,637 a secondary education or less. We included education as a fixed effect in our linear mixed models, so the reported effects of age, gender, commute duration, and home environment are controlled for the effect of education. Similarly, the variance of the random intercepts clustered by countries used for the cross-country analysis is computed after taking into account the variance due to the fixed effects, including education. Also, the pattern and the change points of the sleep duration as a function of age are the same for participants with higher and lower education. We ran the same analysis as presented in Fig. 2B (Wayfinding performance as a function of reported sleep duration) with participants stratified by education level and gender.

We added Figure S6 and the following paragraph to the results (page 5):

"The optimal wayfinding performance was achieved by participants who reported sleeping 7h in all age groups. However the association between reported sleep duration and wayfinding performance was most significant for 54 - 70 yrs participants, and the inverted U-shaped pattern was stronger in this last age group, see Fig S5. This pattern also held when stratifying participants by gender and level of education, see Fig S6."

Fig. S6 Association between reported sleep duration and spatial ability in the 3 age groups identified in Fig. 1, stratified by level of education (top) and gender (bottom). Spatial ability was quantified with the Sea Hero Quest (SHQ) wayfinding task. Training levels did not require spatial ability as the target was visible from the starting point. Error bars correspond to standard errors.

Minor:

- Figures:

- o pg. 3/16: Figure 1B:

- o caption units should be reported in hours, not years.
corrected.

- o WEIRD is used in caption/figure but has not yet been defined.

- o We added the definition in the caption.

- o Fig 3A: part of Alaska is cut off on the country clusters map

- o We removed to gain space in the image improving the legibility of the rest of the map.

- WEIRD- Please check definition of - pg 8 (“WEIRD stands for Rich, Educated, Industrialized, Rich and Democratic”) and actually this should be defined much earlier in paper, before it’s used in Figure 1)

- o In addition to the acronym definition in Fig1, we added the following paragraph page 8:

- o “Henrich and colleagues showed that people from WEIRD (Western, Educated, Industrialized, Rich and Democratic) countries represent as much as 80% of participants in behavioral science studies, but only 12% of the world population [23]. To test whether our findings generalize to the populations less represented in the literature, we compared the average reported sleep duration between WEIRD (19 countries, N=545,639) and non-WEIRD countries (44 countries, N=206,130), labeled in Table S1.”

- p 8/16: I’m not sure this sentence is accurate: “Notably, the specific change points we observed in the overall population were exactly matched across WEIRD and non-WEIRD populations, and across men and women.” because on p. 7 the authors write: “change points at 33 yrs and 55 yrs for the WEIRD population and 31 and 52 yrs for the non-WEIRD population, see Fig 1F” maybe they weren’t statistically different?

- o Thank you for spotting this mistake. We modified the sentence in the discussion (page 8):

- o “Notably, the specific change points we observed in the overall population were exactly matched across and across men and women, and very similar in the non-WEIRD populations (31 and 52 yrs). Thus, this pattern of reported sleep duration as a function of age appears robust across various populations.”

- Methods: How is home environment measured? Please define.

- o As for the other demographics, the home environment is self-reported.

- o We added a citation to our recent paper specifically dedicated to the association between home environment and spatial ability:

- o [20] Coutrot, A. et al. Entropy of city street networks linked to future spatial navigation ability. Nature 604 (7904), 104–110 (2022) .

- Some more information could be included about how the participants accessed the program. I know it is available on apple devices, but more information should be included here (it matters as some countries may have differences in accessibility to the program, and this should be discussed).

- o We added the following paragraph (page 10):

- o “The Sea Hero Quest app was released on May 2016 on the App Store for iOS and on Google Play for Androids. It was available to the general public in 17 languages: English, French, German, Spanish, Macedonian, Greek, Croatian, Dutch, Albanian, Hungarian, Romanian, Slovak, Czech, Polish, Portuguese, Italian, and Serbian.”

Reviewer #3 (Remarks to the Author):

In the interesting study by Coutrot and colleagues, the authors analyzed subjective sleep duration from a large cohort of participants across 63 countries and they identified three distinct phases in the adult human life course; within the late adulthood (> 54 yrs to 70 yrs) the authors identified an inverted U-shape regulation between sleep duration and cognitive performance as measured by spatial navigation with a peak of optimal performance observed at 7 hours of sleep.

This study has definitively major strengths including the large samples size and the generalizability of the findings across several countries in the world. In addition, the merit of the study is that it highlights the negative effects of long sleep duration on cognitive performance in a specific age group (late adulthood) which can address future research to better understand within this age group at higher risk for cognitive impairment, potential mechanisms for the observed association. The authors might want to highlight in the discussion how their findings can impact future research in the field of sleep and cognition.

We added the following paragraph (page 10):

“Crossing the geographical, linguistic, and cultural boundaries to generalize empirical evidence on sleep has been curbed by methodological difficulties. Collecting a large amount of sleep-related data in a standardized and reliable manner from multiple geographical regions across the world remains a challenge. The joint collection of sleep data with behavioral data is even more challenging, albeit essential to understand the associations between sleep, cognition, and behavior. The current study shows that the spread of video games across the planet can be harnessed by sleep researchers to access larger and more diverse populations, allowing them to replicate findings and make discoveries more likely to be valid at a global scale.”

These findings are in line with the growing body of epidemiological studies showing that prolonged sleep is statistically associated with significant morbidity as well as mortality. In many studies, this association was even stronger than that found with short sleep.

As discussed by the authors, a proposed hypothesis is that depression (either as a confounder or as a causal intermediate) may explain the statistical association between long sleep and poor health. Comorbid sleep disorders, low socioeconomic status, preclinical medical disease, and a high level of inflammation have also been proposed as potential explanations for the association between long sleep and elevated mortality risk.

Unfortunately, with the exception of the economic status, the other proposed mediators have not been taken into account by this study and the authors might want to address them as limitations of their study.

Among the factors that could have significantly affected the findings, there are also the time of the day when the cognitive test was performed (circadian component) and light levels. In fact, both these factors can significantly impact alertness and cognitive performance. The authors should address these as possible limitations in the discussion. ¶

Thank you for these very good points. We added the following paragraph (page 9):

“In this study, we collected and analyzed self-reported sleep durations. Self-reported sleep durations may differ from actual sleep patterns and the concordance may be modulated by age, sex [43], and sleep disorders [44]. Since participant demographics and sleep disorders are also associated with spatial ability [19, 45], it will be important to validate our results with objective sleep parameters recorded using polysomnography. We controlled for the effect of age, gender, level of education, commute duration, home environment and country on sleep duration. Other participant-related variables such as ethnicity, socioeconomic status, neurological and psychiatric disorders, and a high level of inflammation also have been associated with sleep duration variation [5, 7, 46, 47]. External variables such as the day of the week, the time of the day, and light levels when the cognitive test was performed significantly impact sleep duration and cognitive performance [10, 48, 49]. We unfortunately did not collect these variables in our current dataset, but they would be important to include in a future model. Future studies should also assess sleep quality known to be confounded with sleep duration.”

There are aspects of the manuscript that might be improved to favor clarity and readability:

- Can the authors explain what age² and education² stand for?

We added the following sentence (page 4):

“A linear mixed model (LMM) was calculated to predict reported sleep duration with age, age² (the quadratic component of age, since Fig. 1 shows a U-shaped curve), gender, education, home environment, commute duration and their interaction with age as fixed effects, and random intercepts clustered by countries”

and page 5:

“(the quadratic component of sleep duration, since Fig. 2B shows an inverted U-shaped curve)”

- Can the author simplify the definition of what wayfinding performance represents?

We added the following sentence (page 13):

“To sum up, WF and TP vary in the opposite direction than the trajectory length. Short, thus optimal trajectories, correspond to a high performance. Positive WF and TP values correspond to above average performance, while negative values correspond to below average performance.”

- In the method section, It is not clear how many times the navigation task was performed by participants, please clarify.

We added the following sentence (page 11):

"To provide a reliable estimate of spatial navigation ability, we examined the data from a subset of participants who completed the tutorial levels at least the first four wayfinding levels (up to level 11 in the game, N = 418,152). We only analysed the first attempt of each level. When the participant retried a level to improve their score, we did not include the associated trajectory in the analysis."

- The authors state that they remove participants > 70 yrs old as they have a strong selection bias: can the authors better explain the nature of the selection bias?

We added the following paragraph (page 2):

"We removed participants above 70 years old because we have previously shown a strong selection bias in participants above this age, causing their performance to be substantially higher than would be expected in unselected participants of the same age [18]. This selection bias was likely due to the fact that people above 70 playing video games on a smartphone are less representative of their class of age, internet and mobile devices adoption rate being particularly low in less educated older adults [22]."

- Looking at Figure 2, a U shape pattern is also detectable for young and middle-aged adults; did the author check for a linear trend across age groups? is it possible that the aging process and the inter-individual variability associated with aging in many physiological processes (sleep and cognition included) makes the U pattern more evident in older adults. Please, expand on this.

Thank you for this suggestion. As we stated on page 5, "the interaction of both the linear and quadratic terms of reported sleep duration with the age groups are significant for wayfinding but not training performance (see Table S3)."

To further examine this trend we plotted the quadratic sleep duration coefficients from 3 ANOVAs independently computed for each age group (Fig S5). While the quadratic sleep duration coefficient for the training performance remains at the same level, the absolute value of this coefficient for the wayfinding performance increases with age, indicating that the inverted U pattern is stronger in older adults.

We have now added the following Fig. S5 and paragraph (page 5)

"The optimal wayfinding performance was achieved by participants who reported sleeping 7h in all age groups. However the association between reported sleep duration and wayfinding performance was most significant for 54 - 70 yrs participants, and the inverted U-shaped pattern was stronger in this last age group, see Fig S5. This pattern also held when stratifying participants by gender and level of education, see Fig S6."

Fig. S5 Quadratic sleep duration coefficients from 3 ANOVAs independently computed for each age group identified in Fig. 1, with Wayfinding Performance and Training Performance as the response variable. The structure of the tested models is "performance ~ sleepduration + sleepduration² + education + home environment + commute duration". Error bars correspond to 95% confidence intervals.

REVIEWERS' COMMENTS

Reviewer #1 (Remarks to the Author):

The authors successfully addressed my comments. Thank you for the invitation to review this interesting manuscript. I have no further comments for the authors.

Reviewer #2 (Remarks to the Author):

The authors have completed a very thorough revision which highlights the study limitations adequately and addresses my main concerns regarding population bias. Regardless of any minor imitations, the study is very exciting and presents unique data.

Reviewer #3 (Remarks to the Author):

The reviewer thanks the authors for carefully addressing all the comments.

Response to the reviewers' comments

Manuscript number: NCOMMS-22-10397

Author names: Coutrot A., Lazar A. S., Richards M., Manley E., Wiener J. M., Dalton R. C., Hornberger M. and Spiers H. J.

Title: Reported sleep duration segments the adult life-course into three phases

We thank all reviewers for their helpful assessment of our manuscript.

REVIEWERS' COMMENTS

Reviewer #1 (Remarks to the Author):

The authors successfully addressed my comments. Thank you for the invitation to review this interesting manuscript. I have no further comments for the authors.

Reviewer #2 (Remarks to the Author):

The authors have completed a very thorough revision which highlights the study limitations adequately and addresses my main concerns regarding population bias. Regardless of any minor imitations, the study is very exciting and presents unique data.

Reviewer #3 (Remarks to the Author):

The reviewer thanks the authors for carefully addressing all the comments.